# Effective La-Na Co-Doped TiO_2_ Nano-Particles for Dye Adsorption: Synthesis, Characterization and Study on Adsorption Kinetics

**DOI:** 10.3390/nano9030400

**Published:** 2019-03-09

**Authors:** Inderjeet Singh, Balaji Birajdar

**Affiliations:** Special Centre for Nano Sciences, Jawaharlal Nehru University, New Delhi 110067, India; lathaphysics69@gmail.com

**Keywords:** La-Na co-doped TiO_2_, non-aqueous solvent controlled sol-gel route, physical adsorption, methylene, blue

## Abstract

The mesoporous La-Na co-doped TiO_2_ nanoparticles (NPs) have been synthesized by non-aqueous, solvent-controlled, sol-gel route. The substitutional doping of large sized Na^+1^ and La^+3^ at Ti^4+^ is confirmed by X-ray diffraction (XRD) and further supported by Transmission Electron Microscopy (TEM) and X-ray Photo-electron Spectroscopy (XPS). The consequent increase in adsorbed hydroxyl groups at surface of La-Na co-doped TiO_2_ results in decrease in pH_IEP_, which makes nanoparticle surface more prone to cationic methylene blue (MB) dye adsorption. The MB dye removal was examined by different metal doping, pH, contact time, NPs dose, initial dye concentration and temperature. Maximum dye removal percentage was achieved at pH 7.0. The kinetic analysis suggests adsorption dynamics is best described by pseudo second-order kinetic model. Langmuir adsorption isotherm studies revealed endothermic monolayer adsorption of Methylene Blue dye.

## 1. Introduction

A leading source of water pollution is dye containing effluents from industries like textile [1], printing [2], leather [3], pharmaceuticals [4] and kraft bleaching [5] and so forth. Many of these dyes are mutagenic, carcinogenic and even lead to chromosomal fractures [6,7,8] causing health hazards to living beings. Eliminating such dyes from industrial effluents is becoming increasingly necessary. Prominent methods used for this are: adsorption [9,10,11], coagulation/flocculation [12,13], membrane filtration [14,15] and so forth. Among these, adsorption methods are advantageous because they are simple, economical and effective. In addition, adsorption efficiency can be controlled by many factors including adsorbent surface area, adsorbent dose, pH, contact time and adsorbate concentration [16,17,18].

Recently nanostructured material such as TiO_2_ [19,20], magnetic iron oxides [21,22,23] and nanoparticle loaded carbon [24,25] have been reported by several researchers for efficient removal of organic dyes. Due to their large surface area and easy tailoring of surface properties, use of nanomaterials can be very effective in adsorbing dyes from industrial effluents. The pH_IEP_ (the pH at which zeta potential of nanoparticle equals zero) plays a vital role in the adsorption kinetics of dye at nanoparticle surface because it affects formation of bonds between surface hydroxyl groups and dye molecules during chemisorption. Hence, modification of nanoparticle surface can be a good strategy for physical removal of organic dye. TiO_2_ is well studied material, which allows easy tailoring of its surface by doping and varying synthesis methods. Alkali dopants [26,27,28,29,30,31] have been long used to enhance surface area of TiO_2_ nanoparticles and creation of new surface sites which, significantly improve the adsorption of dyes. On the other hand, it is reported that minute doping of rare earth elements in TiO_2_ [32,33,34,35] not only concentrate the organic dye at nanoparticle surface but also stabilizes the meso-structure of TiO_2_.

Therefore modification of TiO_2_ using co-doping of rare earth elements and alkali metals is expected to be more effective in the adsorption of dye at the surface. In view of this, pure, La doped, Na doped and La-Na co-doped TiO_2_ nanopowder were prepared by solvent controlled non aqueous sol-gel route [36,37,38]. A systematic investigation was carried out to understand the role of dopants on adsorption property of dye at nanoparticle surface. Earlier reports [26,39] demonstrated that Na^+^ (1.02 Å) and La^+3^ (1.03 Å), due to their large size, could not enter in TiO_2_ lattice to substitute for Ti^4+^ (0.68 Å) but migrates to TiO_2_ surface. In present work, substitutional doping of Na and La is confirmed by XRD, TEM and XPS for the first time. The variation in pH_IEP_ values with doping is well explained and correlated with adsorption behavior of dye at modified TiO_2_ surface.

## 2. Experimental Section

### 2.1. Synthesis Method

Non-aqueous, solvent-controlled, sol-gel route is excellent for preparation of highly pure oxide nanoparticles (NPs) with good yield and is therefore adopted for the synthesis of TiO_2_ NPs. Pure TiO_2_ gel was prepared by mixing 20 mL Titanium tetra isopropoxide (Spectrochem, Mumbai, India) with 40 mL methyl cellosolve (SRL Chem, Mumbai, India) by stirring for 2 h at pH 3 (maintained using 1 M HNO_3_ (Merck, Darmstadt, Germany). Under similar conditions, gels of La doped TiO_2_, Na doped TiO_2_ and La-Na co-doped TiO_2_ were prepared using lanthanum nitrate (SRL, Mumbai, India) and sodium nitrate (CDH, Mumbai, India) as La and Na precursor respectively. The labels and nominal composition of samples are indexed in Table 1. The prepared gels were dried under Infrared IR lamp and grinded to fine powder. For crystallization, all powders were calcined at 450 °C for 1 h. The substantial concentrations of metal doping were determined by wavelength dispersive X-ray fluorescence spectroscopy (WDXRF) and obtained results are also included in Table 1.

### 2.2. Characterization

The concentration of metal dopant was identified by wavelength dispersive X-ray fluorescence spectroscopy (WDXRF) (Bruker S4 PIONEER, Texas, USA). X-ray diffraction (XRD) patterns were acquired on Rigaku diffractometer (Cu K_α_, λ = 0.154 nm) to determine crystallite size. The Brunauer-Emmett-Teller (BET) analysis of NP’s was performed using Quantachrome Instrument, Florida, USA after degassing at 300 °C for 4 h. JEOL 2100-F (Tokyo, Japan) TEM operating at 200 kV was used for transmission electron microscopy (TEM) analysis. Transmission electron microscope (TEM) images, high resolution TEM (HRTEM) images, selective area electron diffraction (SAED) pattern, Scanning TEM (STEM) images & energy dispersive X-ray (EDX) spectra were acquired to determine crystallite size, *d*-spacing, crystallinity and elemental analysis. The binding energy of each element present in sample was determined via X-ray photoelectron spectroscopy (XPS) (XPS oxford instrument using X-rays of energy 1486.6 eV, Concord, MA, USA). Zeta potential analyzer (ZEECOM Microtec, Tokyo, Japan) was used to determine isoelectric point (IEP) of the prepared nanopowder.

### 2.3. Adsorption Studies

Methylene blue (Hi media, Pune, India) dye adsorption at NPs surface was carried out by batch technique. All the adsorption experiments were carried out at room temperature (25 °C). The effect of varying metal doping, initial pH, contact time, adsorbent NPs dose, initial dye concentration and temperature on adsorption was studied. In general, a particular amount of NPs dose is added in 10 ml methylene blue (MB) dye solution and stirred for required time. The pH of the solution is adjusted by using 0.01 M NaOH and 0.01 M HCl. The adsorption capacity *q_e_* (mg/g) and removal percentage of prepared NPs for MB dye were calculated by the Equations (1) and (2) [25]:*q_e_* = (*C*_0_ − *C_e_*)*V*/*m*(1)
*Removal* % = ((*C*_0_ − *C_t_*)/*C*_0_) × 100(2)
where, *C*_0_ (mg/L), *C_e_* (mg/L) and *C_t_* (mg/L) are the MB dye concentrations at initial time, equilibrium time and contact time *‘t’* respectively. *V* (L) is the total volume of the suspension and m (g) is mass of adsorbent NPs. In order to measure concentration of dye at different contact time, NPs are removed from the solution by centrifugation and absorbance of supernatant measured by UV-Visible spectrometer. These absorbance results are calibrated with standard samples and then concentration of residual dye is measured. For data consistency, all the adsorption experiments are performed in duplicates.

## 3. Results and Discussion

### 3.1. XRD Analysis

Structural characterization of nanopowder was carried out by XRD and is reported in Figure 1. XRD patterns of PT show the diffraction peaks of pure TiO_2_ anatase phase. Similar peaks are observed for LT, NT4, LNT2, LNT4 and LNT6. Absence of extra peaks confirms the formation of pure anatase phase in doped NPs and that there are no segregated phases of dopants. The shifts in peak position with doping confirm the substitutional doping of large sized La^+3^ (1.03 Å) and Na^+1^ (1.02 Å) dopant at Ti^4+^ (0.68 Å) site. Generally, doping of large ionic radii metal ions in host lattice TiO_2_ results in strain in crystal structure. This lattice strain is compensated by formation of oxygen vacancies [38,40]. Secondly the substitution of host lattice ion by low valence metal ion is also likely to yield oxygen vacancies [41]. Thus, doping of large sized and low valent Na or La metal ions in TiO_2_ matrix would induce formation of oxygen vacancies. Besides, La and Na mono-doping and co-doping in TiO_2_ reduces the intensity of X-ray peaks of anatase phase. As compared to Na doping, La doping shows stronger reduction in the intensity of anatase phase. This could be attributed to fewer oxygen vacancies expected upon La doping as compared to Na due to high valence state of La.

The crystal plane (101) was selected to determine average crystallite size of prepared nano-powders using Debye Scherrer formula. The calculated average crystallite size and 2θ values are tabulated in Table 1. Clearly, the average crystallite size decreases for individual as well as co-doped TiO_2_ nano-powder. Besides, there is critical doping concentration (LNT4) after which crystallite size increases. The decrease in crystallite size is obviously due to substitution of low valent and large sized dopants resulting in strained crystal structure and hence oxygen vacancies [41,42].

### 3.2. BET Surface Areas and Pore Size

The mesoporous structure and surface area of prepared samples were examined by isothermal curves formed by adsorption and desorption of N_2_ and the Barrett-Joyner-Halenda (BJH) pore size distribution curves. As shown in Figure 2a, all prepared NPs exhibited H2 type hysteresis loop with adsorption-desorption isotherms of type 4, typical characteristic of mesoporous structure with ink-bottle shaped pores [43,44]. The respective pore size distribution curves (Figure 2b) are obtained from desorption isotherm of pure and doped TiO_2_ samples using BJH method. The BET surface area, pore volume and pore diameter of all the samples are tabulated in Table 2.

Clearly, there is variation in hysteresis loop and pore distribution curve with doping. Increase in La and Na doping in pure TiO_2_ results in large surface area due to decreased crystallite size. In addition, pore size and hence pore volume increases with certain level of dopant concentration as in LNT4 and thereafter decreases. This could be attributed to formation of oxygen vacancies [45]. Moreover, this speculation is also supported by high porosity of NT4 sample as compared to LT, because NT4 possess more oxygen vacancies.

### 3.3. Morphology

TEM, HRTEM and SAED give structural and morphological information about prepared nanopowder. TEM bright field images of Figure 3a PT and Figure 3b LNT4 further corroborate XRD result that crystallite size of LNT4 is smaller than PT nanopowder. The representative particle size of LNT 4 nanopowder (8–12 nm) is smaller than PT nanopowder (15–20 nm). In addition, SAED pattern (Inset of Figure 3a,b) form ring patterns which are indexed according to anatase phase of TiO_2_ which confirm the crystalline phase of PT and LNT4. The substitutional doping of dopants is further confirmed by increased d spacing in LNT4 (Figure 3d) as compared to that of PT (Figure 3c).

Figure 3e shows STEM dark field image of cluster of LNT4 nanopowder. The simultaneous presence of La, Na and Ti peak in point EDX spectrum (Figure 3f) taken from the area shown by a black circle (Figure 3e) corroborate XRD and HRTEM results.

### 3.4. XPS Analysis

The binding energy of different elements present in prepared samples, were examined by XPS analysis. The obtained spectra were baseline corrected and then fitted by commonly used Voigt function. As shown in Figure 4, the high resolution O 1s XPS spectra of all samples composed of two peaks corresponding to Ti-O link and adsorbed hydroxyl groups (Ti–OH link) [46]. The increase in peak intensity of Ti-OH link with Na and La doping confirms increased adsorption of hydroxyl groups at the surface. This is due to increased formation of oxygen vacancies [47] at the surface of nanoparticle due to substitutional doping of La and Na at Ti site (confirmed by XRD). Furthermore, substitutional doping of La and Na at Ti site is also confirmed by comparative study of high resolution XPS spectra of Ti 2p (Appendix A). The doublet of Ti 2p of PT is composed of Ti 2p_1/2_ (B.E. 464.4 eV) and Ti 2p_3/2_ (B.E. 458.7 eV) and indicates Ti^+4^ state. The binding energy shows red shift (Table 3) with La and Na doping, which could be attributed to the lower electronegativity of La (1.10) and Na (0.93) than that of Ti (1.52). This result confirms substitutional doping of La and Na at Ti site [48] and absence of other oxidation state of Ti.

In addition, the high resolution XPS spectra of La 3d (Appendix A) and Na 1s (Appendix A) confirm the simultaneous presence of La and Na. The La 3d peak composed of multiplet 3d_5/2_ and 3d_3/2_ with core level binding energy 834.9 eV and 851.8 eV respectively corresponding to La^+3^ oxidation state [49]. The distance between two peaks La3d_5/2_ (peak 1) and La 3d_3/2_ (peak 1) corresponds to bonding of La with O [50]. Similarly, Na 1s peak with core level binding energy 1071.3 eV represents Na^+1^ oxidation state [51].

### 3.5. Zeta Potential Study and Isoelectric Point

The surface charge of nanoparticles play a vital role on adsorption kinetics of dye at surface of nanoparticles. Zeta potential measurement technique was used to determine surface charge. The pH value at which there is no charge at the surface and hence zero zeta potential is defined as isoelectric point. The surface of suspended TiO_2_ nanoparticles in water covered with hydroxyl groups [52] and therefore surface charge is function of pH of solution according to Equations (3) and (4),
Ti^+4^–OH + H^+^ → Ti^+4^–OH_2_^+^(3)
Ti^+4^–OH → Ti^+4^–O^−^ + H^+^(4)

When pH of solution is less than pH_IEP_, nanoparticle surface is positively charged according to Equation (3) and favors adsorption of anionic species. Whereas, when pH of solution is greater than pH_IEP_, nanoparticle surface is negatively charged according to Equation (4) resulting in adsorption of cationic species.

Figure 5A shows the zeta potential values of different nanopowder as function of pH of solution. It is clear from Figure 5B that pH_IEP_ value decreases from PT to LNT6. This is attributed to increased surface area and adsorbed hydroxyl groups [45,52]. The increased surface area results in more adsorption of hydroxyl groups and hence more hydrogen ions are produced, which decreases pH_IEP_ value [53]. A higher concentration of hydroxyl groups at the surface of nanoparticle is expected to favor more adsorption of dye molecules.

### 3.6. Adsorption Studies

Effect of metal doping. The effect La and Na doping in TiO_2_ on adsorption of MB dye was studied at optimal conditions [equilibrium time = 15 min; pH = 7.0; dye conc. = 5 mg/L; NPs dose = 1.2 g/L] by considering absorbance peak at 664 nm. At pH 7, all NPs have negative zeta potential (Figure 5A) which facilitates adsorption of cationic MB dye. It is clear from Figure 6a that LNT4 NPs shows best removal rate of MB dye, even if LNT2, LNT4 and LNT6 possess nearly equal negative zeta potential. This is attributed to small crystallite size and large surface area of LNT4 as compared to LNT2 and LNT6.

Effect of initial pH. The LNT4 NPs shows best dye removal percentage among all prepared NPs. Therefore, further adsorption experiments are performed using LNT4 NPs as adsorbate. The absorbance spectra of MB dye at different pH is shown in Appendix A. There is no change in peak position and shape of absorbance spectra observed. Figure 6b shows effect of pH on dye removal percentage by LNT4 NPs [equilibrium time = 15 min; dye conc. =5 mg/L; NPs dose = 1.2 g/L]. At lower pH, LNT4 shows very less dye removal percentage, which is obvious due to positive zeta potential resulting in repulsive force on cationic dye. However, when pH is greater than pH_IEP_, dye removal percentage increases. For example, at pH 7.0, 99% of the dye is removed. This is because of electrical attraction between cationic MB dye and negative surface charge of LNT4 at pH > pH_IEP_. At pH more than 7.0 there is negligible increase in removal percentage. Therefore, all further sorption experiments were conducted at pH 7.0.

Effect of contact time. The effect of contact time on dye adsorption at LNT4 NPs surface was investigated at different time intervals (2, 5, 10, 15, 20 and 25 min) at optimal conditions [pH = 7.0; dye conc. =5 mg/L; NPs dose = 1.2 g/L]. Dye removal percentage increased from 47% to 99% with increase in contact time from 2 min to 25 min. The sorption equilibrium was achieved within 15 min. Attaining equilibrium in such short span of time indicates chemical adsorption of dye, instead of physical adsorption that takes relatively long contact time [54]. After 15 min, there is negligible uptake of dye is observed (Figure 6c). Therefore, further adsorption experiments were carried out for 15 min.

Effect of LNT4 NPs dose. The effect of LNT4 NPs dose on MB dye uptake is shown in Figure 6d. MB dye sorption experiments were conducted using 5 mg/L dye solution at pH 7.0 [equilibrium time = 15 min]. The LNT4 NPs dose varied from 0.6 g/L to 1.6 g/L. MB dye removal percentage increased considerably from 73% to 99% on increasing dose from 0.6 g/L to 1.2 g/L. This increase in removal percentage with increase in adsorbent dose is obvious and attributed more available surface area and active surface sites [55]. No further uptake of MB dye was observed on increasing NPs dose to 1.4 g/L and 1.6 g/L. Therefore, 1.2 g/L dose was chosen as optimum dose for further adsorption experiments.

Effect of initial dye concentration. The effect of initial dye concentration on dye sorption was investigated using 5–15 mg/L MB dye concentration [equilibrium time =15 min; pH =7.0; NPs dose =1.2 g/L]. The dye removal percentage drops from 99% to 42% as dye concentration is increased from 5 mg/L to 15 mg/L (Figure 6e) due to less available surface area and surface sites. Similar results were reported by many authors [16,56,57].

Effect of temperature. Generally wastewater effluents from industries are hot and hence adsorbents used must be stable and thermal resistant [58]. Therefore, investigation of dye adsorption at different temperature is important. Figure 6f shows the removal rate of MB dye at different temperature by LNT4 NPs [equilibrium time =15 min; pH =7.0; NPs dose =1.2 g/L; dye conc.= 5 mg/L]. Clearly there is more adsorption of MB dye with rise in temperature, which implies endothermic adsorption process. The favorable increase in adsorption of dye with rise in temperature can be explained by two ways [10]: (i) elevated interaction between cationic MB dye and hydroxyl groups at surface of NPs; (ii) enhanced diffusion of MB dye molecules within pores of NPs. In addition, there may be disruption of agglomeration with rise in temperature, which facilitates increase in surface area and hence increased dye adsorption. Many reports [57,59] investigated effect of temperature on adsorption of dye at adsorbent surface and similar results were found.

### 3.7. Adsorption Kinetics

In order to investigate the mechanism of adsorption, several kinetic models are used to fit experimental data. Most commonly used kinetic models, namely pseudo first order, pseudo second order and intra-particle diffusion are used to determine preferential kinetic model for dye adsorption. The best fit model is determined by comparing linear regression correlation coefficient (R^2^) value. The simple form of pseudo-first-order, pseudo-second-order and intra-particle diffusion model might be represented by Equations (5)–(7) respectively.
*log* (*q_e_* − *q_t_*) = *log* (*q_e_*) − (*k*_1_/2.303)*t*(5)
(*t*/*q_t_*) = 1/*k*_2_*q_e_*^2^ + *t*/*q_e_*(6)
*q_t_* = *k_i_t*^1/2^ + *C*(7)
where *k*_1_ (min^−1^), *k*_2_ (g mg^−1^ min^−1^) and *k_i_* (mg g^−1^ min^−1/2^) are rate constants of pseudo-first-order, pseudo-second-order reaction and intra-particle diffusion respectively and *C* is the intercept. *q_e_* and *q_t_* are the adsorption capacity at equilibrium and at contact time *‘t’* respectively. The rate constant k_1_ could be determined from slope of linear plot log (*q_e_* − *q_t_*) versus *t*. Similarly, slope of linear plot of *t/q_t_* versus t gives the rate constant *k*_2_. The rate constant *k_i_* can be evaluated from the slope of the plot of *q_t_* versus *t*^1/2^.

The plot of pseudo first order, second order and intra-particle diffusion kinetic model obtained for adsorption of MB dye at the surface of LNT4 NPs is shown in Figure 7A–C. The obtained regression coefficient (*R*^2^) and calculated k values are listed in Table 4. The value of regression coefficient (*R*^2^) of second order kinetic model is higher than first order, which could be attributed to heterogeneous nature of TiO_2_ NPs [23]. Thus the kinetics of MB dye adsorption on LNT4 NPs is best described by pseudo second order kinetic model.

The slope of intra-particle diffusion plot measure rate of intra-particle diffusion, which can occur after external surface adsorption. Whereas, intercept of plot is proportional to boundary layer thickness and measure the contribution of surface adsorption to rate controlling step [60]. The low value of regression coefficient (Figure 7c) reflects less involvement of intra-particle diffusion to rate controlling step.

### 3.8. Adsorption Isotherm

The binding mechanism of dye at NPs surface can be understood by analyzing adsorption isotherm. The adsorption isotherm of MB dye at surface of NPs was studied using Langmuir model, which consider adsorbent surface to be mono-layered with uniform energy at which adsorption occurs [21]. The Langmuir isotherm equation is given by Equation (8) [61] as:*C_e_*/*q_e_* = 1/*bq_max_* + *C_e_/q_max_*(8)
where, *b* (L/mg) is Langmuir isotherm adsorption constant and *q_max_* (mg/g) is adsorption capacity, which are calculated using slope and intercept, obtained from linear plot of *C_e_/q_e_* versus *C_e_* (Figure 8). 

The Langmuir adsorption constants and correlation coefficient (*R*^2^) for adsorption of MB dye at two different temperatures are given in Table 5. The value of *R*^2^ is higher than 0.9, which confirms that monolayer adsorption of dye is predominant. In addition, the value of *q_max_* calculated by Langmuir isotherm model increases with rise in temperature indicating adsorption process is endothermic.

## 4. Conclusions

Physical removal of organic cationic MB dye by La-Na co-doped TiO_2_ nano-powder prepared by non-aqueous, solvent-controlled, sol-gel route is demonstrated. The prepared NPs are crystalline and mesoporous as confirmed by XRD and BET analysis. Furthermore, XRD and TEM results confirm the substitution of large sized Na^+1^ and La^+3^ at Ti^+4^ sites. This low valent metal ion doping results in formation of oxygen vacancies which facilitates more adsorption of hydroxyl groups at the surface of NPs (confirmed by XPS). The adsorbed hydroxyl groups reduce the pH_IEP_ value and, therefore facilitate effective adsorption of cationic MB dye. The adsorption capacity of LNT4 NPs is found to be highest, which could be attributed to its high surface area and porosity. In addition, adsorption kinetics of MB dye at surface of LNT4 NPs is best described by pseudo second-order kinetic model due to heterogeneous nature of titania NPs. Langmuir adsorption isotherm studies revealed endothermic monolayer adsorption of Methylene Blue dye.

## Figures and Tables

**Figure 1 nanomaterials-09-00400-f001:**
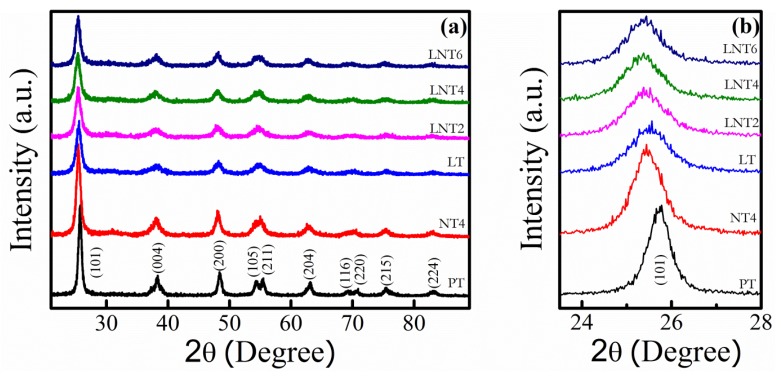
(**a**) XRD patterns of NPs. (**b**) Amplified image of (**a**).

**Figure 2 nanomaterials-09-00400-f002:**
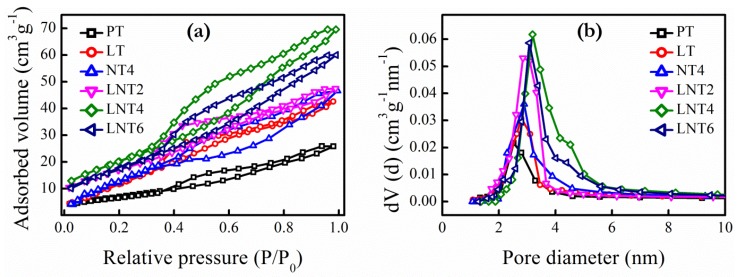
(**a**) N_2_ adsorption-desorption isotherms and (**b**) pore diameter distribution curve of prepared NPs.

**Figure 3 nanomaterials-09-00400-f003:**
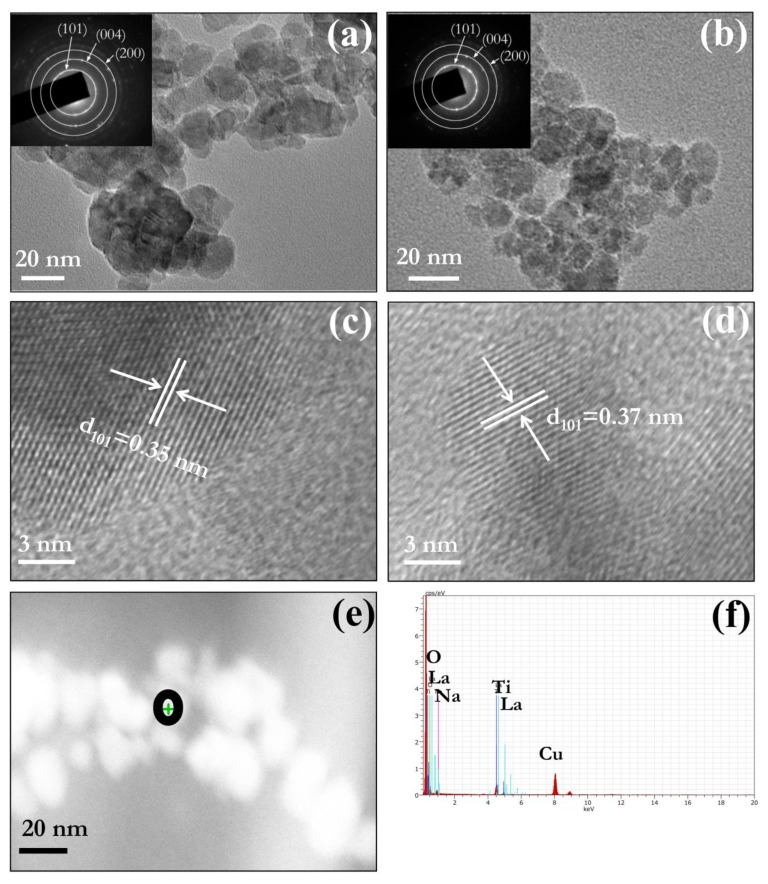
TEM micrograph of (**a**) PT and (**b**) LNT4 nanopowder. Insets in (**a**,**b**) show SAED patterns of PT and LNT4 respectively. HRTEM images of (**c**) PT and (**d**) LNT4. (**e**) STEM dark field image of cluster of LNT4 nanopowder. (**f**) STEM-EDX point spectrum from an area shown by black circle in (**e**).

**Figure 4 nanomaterials-09-00400-f004:**
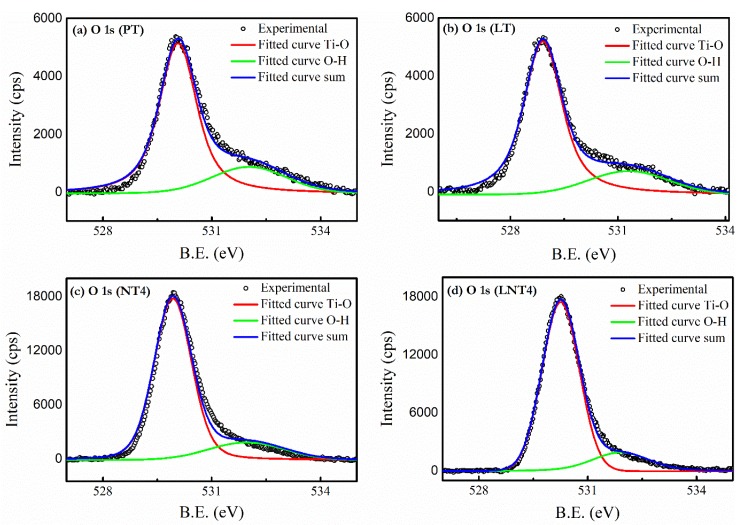
The High resolution XPS spectra of O 1s for (**a**) PT, (**b**) LT, (**c**) NT4 and (**d**) LNT4.

**Figure 5 nanomaterials-09-00400-f005:**
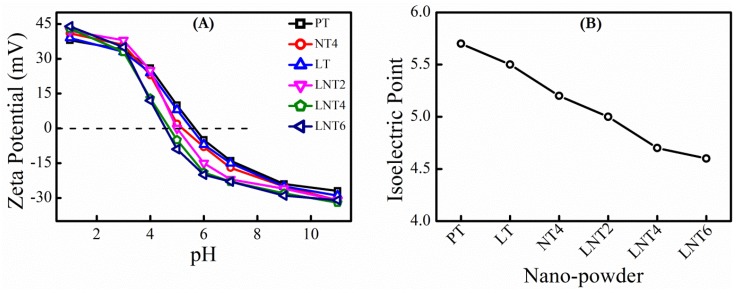
(**A**) Effect of doping in TiO_2_ on dispersion zeta potential at different potential. (**B**) The dispersion isoelectronic point (IEP) for different NPs.

**Figure 6 nanomaterials-09-00400-f006:**
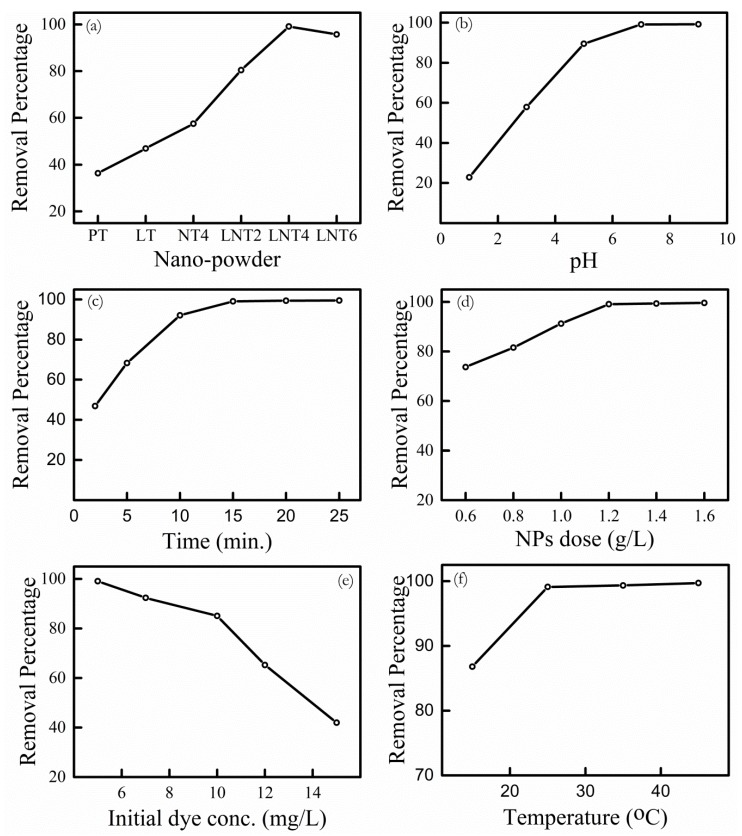
(**a**) Effect of metal doping in TiO_2_ on dye removal. (**b**–**e**) Dye removal by adsorption for LNT4: (**b**) Effect of solution pH. (**c**) Effect of contact time. (**d**) Effect of LNT4 NPs dose. (**e**) Effect of initial dye concentration. (**f**) Effect of temperature.

**Figure 7 nanomaterials-09-00400-f007:**
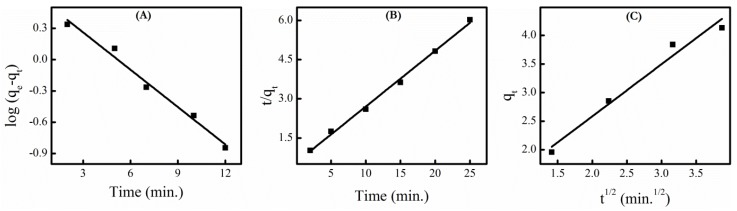
(**A**) pseudo first order, (**B**) pseudo second order and (**C**) intra-particle diffusion kinetic plots for adsorption of dye at LNT4 NPs surface.

**Figure 8 nanomaterials-09-00400-f008:**
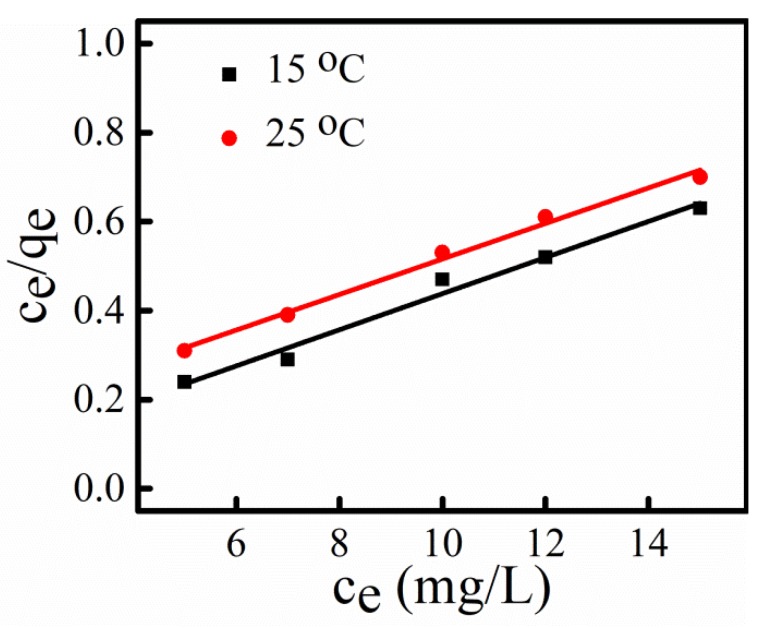
Langmuir isotherm plot for adsorption of MB dye at two different temperatures.

**Table 1 nanomaterials-09-00400-t001:** Nominal & actual dopant concentration, 2θ values and crystallite size of NPs.

NPs Labelling	Nominal Dopant Conc. (at. %)		Dopant Conc. (at. %)		2θ (Degree) A(101)	Crystallite Size (nm)
	Na	La	Na	La		
PT	0	0	0	0	25.722	14.0
LT	0	1	0	0.90	25.512	8.0
NT4	4	0	3.89	0	25.466	10.0
LNT2	2	1	1.82	0.81	25.425	7.5
LNT4	4	1	3.75	0.76	25.359	7.0
LNT6	6	1	5.67	0.74	25.380	9.0

**Table 2 nanomaterials-09-00400-t002:** The pH_IEP_, BET surface area, BJH pore volume and pore diameter of all synthesized NPs.

NPs	pH_IEP_	BET Surface Area (m^2^ g^−1^)	Pore Volume (cm^3^ g^−1^)	Pore Diameter (nm)
PT	5.7	45.712	0.039	2.32
LT	5.5	54.342	0.052	2.59
NT4	5.2	49.629	0.066	2.70
LNT2	5.0	63.418	0.080	2.85
LNT4	4.7	74.996	0.119	3.19
LNT6	4.6	67.971	0.104	3.08

**Table 3 nanomaterials-09-00400-t003:** Peak position of different peaks obtained from high resolution XPS of different samples.

Sample	Peak Position								
	Ti-O Peak	O-H Peak	Ti 2p_3/__2_ Peak	Ti 2p_1/2_ Peak	La 3d_5/2_		La 3d_3/2_		Na 1s Peak
					Peak 1	Peak 2	Peak 1	Peak 2	
PT	530.07	532.03	458.71	464.41	-	-	-	-	-
LT	529.93	531.96	458.46	464.16	834.99	839.18	851.51	856.04	-
NT4	528.91	531.29	458.19	464.06	-	-	-	-	1071.34
LNT4	529.90	531.91	457.75	463.38	834.83	839.03	851.38	855.89	1071.23

**Table 4 nanomaterials-09-00400-t004:** Adsorption kinetics parameter obtained from adsorption of MB dye at surface of LNT4 NPs.

1st Order Kinetics		2nd Order Kinetics		Intra-Particle Diffusion	
*k* _1_	*R* ^2^	*k* _2_	*R* ^2^	*k_i_*	*R* ^2^
0.253	0.9796	0.081	0.9959	0.762	0.9616

**Table 5 nanomaterials-09-00400-t005:** Adsorption isotherm parameter obtained from adsorption of MB dye at surface of LNT4 NPs.

Langmuir Isotherm Constant	15 °C	25 °C
***q_max_***	24.62	25.04
***b***	1.35	0.33
***R*^2^**	0.97	0.99

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
