# Peer review of "Effective La-Na Co-Doped TiO_2_ Nano-Particles for Dye Adsorption: Synthesis, Characterization and Study on Adsorption Kinetics"

_nanomaterials, 2019, doi:10.3390/nano9030400_

Reviewer 1 Report

Dear Editor,

The manuscript (nanomaterials-383554) is about development of an adsorbent system based on doped TiO2 nanoparticles. The study is interesting and extensive and more or less covers different aspects of an adsorbent system’s performance. However, there are still some major concerns that need to be addressed before publication. My major comments are:

1- The English writing language is in general fine but needs a proofreading to minimize grammatical mistakes.

2- The dyeing wastewaters are typically hot (Materials 2016, 9(10),848), while the authors have ignored the role of temperature in their studies. I highly recommend them to investigate the effect of temperature as well.

3- The equations 1, 2 etc. need a reference.

4- Table 1 defining the acronyms of the samples including LT, NT4, LNT2, LNT4 and LNT6 must be included before Results and discussion section. The reader needs to be aware of them as early as possible.

5- How big/small are the nanoparticles? I suggest the authors to determine the particle size and distribution and heterodispersity index.

6- Table 2, please round up/down the values e.g. 3.077nm is weird.

7- TEM micrographs show severe agglomeration of the nanoparticles. Whereas, the nanoparticles are highly negatively charged and must repel each other. Can the authors comment on this effect and how it can be avoided? because certainly agglomeration lowers adsorption efficiency notably.

8- Page 7, the equations are chaotic, thus must be rewritten and renumbered, because there are already eqs. 1 and 2 in page 2.

9- Page 7, lines 186-187: The authors attribute the decline of isoelectric point to increase of surface area thus higher density of OH groups. But, why not composition?

10- Page 7, lines 196-197: The authors attribute the better removal rate of LNT4 to its smaller crystallites and larger surface area compared to LNT2 and LNT6 despite an almost equal zeta potential. But, earlier they claimed that smaller crystallites and larger surface area leads to lower zeta potential!!

11- Page 7, line 201, please convert ppm to mg/L for a better uniformity of the units in the entire text.

12- Fig. 6, are the values reported in graphs average of two data? The authors earlier reported they have performed the experiments two times. If so, can they mention this in the Fig caption.

13- Page 8, 1st and 2nd paragraphs and page 9 1st paragraph, the authors need to discuss the obtained results rather than just reporting their observations.

14- please displace Table 3 into the section of Adsorption Kinetics not before it.

Author Response

Response to reviewer 1

We are thankful to reviewer for reviewing our manuscript.

Comment 1: The English writing language is in general fine but needs a proofreading to minimize grammatical mistakes.

Ans. The English is improved in the revised manuscript.

Comment 2: The dyeing wastewaters are typically hot (Materials 2016, 9(10), 848), while the authors have ignored the role of temperature in their studies. I highly recommend them to investigate the effect of temperature as well.

Ans. As per your valuable suggestion, effect of temperature on dye absorption (Fig. (Figdeen investigated and obser6f) has been investigated and obtained results are included in the revised manuscript.

Comment 3: The equations 1, 2 etc. need a reference.

Ans. The reference for equation 1 and 2 are now included in the revised manuscript.

Comment 4: Table 1 defining the acronyms of the samples including LT, NT4, LNT2, LNT4 and LNT6 must be included before results and discussion section. The reader needs to be aware of them as early as possible.

Ans. Table 1 is now included before results and discussion section in the revised manuscript.

Comment 5: How big/small are the nanoparticles? I suggest the authors to determine the particle size and distribution and hetero-dispersity index.

Ans. Quantitative determination of size distribution and hetero-dispersity index appears difficult because of tendency of nanoparticle to agglomerate. Therefore, at this stage only representative particle size as qualitative measure is given in section 3.3, which is in agreement with average crystallite size determined by XRD in table 1.

 Comment 6: Table 2, please round up/down the values e.g. 3.077 nm is weird.

Ans. The pore diameter values are rounded up/down in the revised manuscript.

Comment 7: TEM micrographs show severe agglomeration of the nanoparticles. Whereas, the nanoparticles are highly negative charged and must repel each other. Can the authors comment on this effect and how it can be avoided? Because certainly agglomeration lowers adsorption efficiency notably.

Ans. The surface negative charge on the nanoparticles revealed by the zeta potential measurement would be relevant only in the suspension state. However, the nano-particles are prone to agglomeration due to hydration solvent loss during or after drying. Therefore, agglomeration likely to occur when nanoparticle suspension is drop-cast on the carbon coated TEM grid. The synthesis method employed does not provide for a separate capping layer to prevent agglomeration which would affect its adsorption properties. Though agglomerated particles exhibit reduced adsorption efficiency, indication of porous structure is clearly seen from the bright-field TEM images especially for LNT4 nano-powders, which supports the mesoporous nature revealed by BET analysis.

Comment 8: Page 7, the equations are chaotic, thus must be rewritten and renumbered, because there are already equation1 and equation 2 in page 2.

Ans. The equations are rewritten and renumbered in the revised manuscript.

Comment 9: Page 7, lines 186-187: the authors attribute the decline of isoelectric point to increase of surface area thus higher density of OH groups. But, why not composition?

Ans. The decline of isoelectric is attributed to increase of surface area thus higher density of OH groups. The increase of surface area, which we have already described in the XRD and BET analysis part, is due to composition. Thus, variation of isoelectric point originates by varying composition.  

Comment 10: Page 7, lines 196-197: The authors attribute the better removal rate of LNT4 to its smaller crystallites and larger surface area compared to LNT2 and LNT6 despite an almost equal zeta potential. But, earlier they claimed that smaller crystallites and larger surface area leads to lower zeta potential!!

Ans. The surface area of LNT4 is much higher than LNT2 and thus pHIEP of LNT4 is lower than that of LNT2. But lower pHIEP of LNT6 as compared to that of LNT4 is still not clear.

Comment 11: Page 7, line 201, and please convert ppm to mg/L for a better uniformity of the units in the entire text.

Ans. The ppm unit is converted into mg/L through-out the revised manuscript.

Comment 12: Fig. 6, are the values reported in graphs average of two data? The authors earlier reported they have performed the experiments two times. If so, can they mention this in the Fig caption.

Ans. The reported graphs in Fig.6 are averaged data so there should be no need to mention them separately using figure caption.

Comment 13: Page 8, 1st and 2nd paragraphs and page 9 1st paragraphs, the authors need to discuss the obtained results rather than just reporting their observations.

Ans. The valuable suggestion is implemented in the revised manuscript.

Comment 14: Please displace Table 3 into the section of Adsorption Kinetics not before it.

Ans. Table 3 is now named as Table 4, which is displaced into the section of adsorption kinetics in the revised manuscript.

Reviewer 2 Report

In this work the authors describe the synthesis of TiO2 nanoparticles doped or co-doped with different concentrations of La and Na by sol-gel technique and their use as adsorbent for MB dye. In particular, their results show that the highest dye removal percentage is achieved by the LNT4 sample (TiO2 np co-doped with La 3.75 at % and Na 0.76 at %) and this property is mainly associated to the high surface area of the nanomaterial.

Overall, the paper is well written and reports interesting results, anyway a few (but important) issues must be addressed before publication.

-        The parameters published in Table 2 for PT and NT4 are different from the values obtained for the (apparently) same materials in a previous paper by the same authors [ref. 39]. Why is there such a difference?

-        XPS results (Page 16, line 155-156): The statement “The increase in peak intensity …. confirms increased adsorption…” needs a quantification, for example it could be useful to add a table with the values of Ti-O and O-H peak areas for O1s obtained in the fitting procedure reported in Fig.4. Furthermore, it is also important to indicate the peak positions and their FWHM, since a red shift for the LT sample is clearly observed, whereas for NT4 and LNT4 samples it is not evident.

-        As concerns the dye concentration measurements: it would be useful to report the MB absorbance spectra obtained at the different pH values as supplementary material. Do the authors observe any change in the absorbance peak position or shape when the pH is varied? It is also required to indicate the wavelenght corresponding to the absorbance peak taken for the concentration calculation.

-        In the Supplementary file, in Fig.S1b, both the Ti2p3/2 and the Ti2p1/2 signals obtained for LT sample seem to be larger (are there more contributions?) than the same peaks for the other samples. How do the authors explain this difference? The fit parameters (position, area and FWHM) for all the samples should be reported in a table and the results should be discussed in the text. The same (a Table with the fitting parameters) should be done for the XPS spectra reported in Fig.S2 and Fig.S3. Indeed the peak positions and the FWHM are important in order to understand the chemical bindings, for example the distance between the two peaks relative to La3d5/2 signal is generally used to identify whether La is bound to O or to OH or to CO3 (La2O3, La(OH3)), La2(CO3)3).

Minor changes:

Page5, line 142 “(Fig.3c)” and “(Fig.3d)” are inverted

Author Response

Response to reviewer 2

We are thankful to reviewer for reviewing our manuscript.

Comment 1:  The parameters published in Table 2 for PT and NT4 are different from the values obtained for the (apparently) same materials in a previous paper by the same authors [ref. 39]. Why is there such a difference?

Ans.  This is due to effect of degassing temperature during BET analysis. In present manuscript the samples were degassed at 300 oC for 4 hour. While in ref. 39 samples were degassed at 400 oC for 2 hour.

Comment 2: XPS results (Page 16, line 155-156): The statement “The increase in peak intensity …. confirms increased adsorption…” needs a quantification, for example it could be useful to add a table with the values of Ti-O and O-H peak areas for O1s obtained in the fitting procedure reported in Fig.4. Furthermore, it is also important to indicate the peak positions and their FWHM, since a red shift for the LT sample is clearly observed, whereas for NT4 and LNT4 samples it is not evident.

Ans. The authors are thankful for this valuable suggestion. The area under Ti-O and O-H peaks is now tabulated in Table S1 and FWHM of different peaks are mentioned in Table S2. The corresponding peak positions are given in Table 3 of revised manuscript. Clearly, there is shift in peak position with doping.

Comment 3:  As concerns the dye concentration measurements: it would be useful to report the MB absorbance spectra obtained at the different pH values as supplementary material. Do the authors observe any change in the absorbance peak position or shape when the pH is varied? It is also required to indicate the wavelength corresponding to the absorbance peak taken for the concentration calculation.

Ans. The absorbance spectra of MB at different pH is shown in Fig. S4 of supplementary file. There is no change in peak position or shape of absorbance peak when pH is varied. The concentration of dye is calculated using absorbance peak at 664 nm.

Comment 4:  In the Supplementary file, in Fig.S1b, both the Ti2p3/2 and the Ti2p1/2 signals obtained for LT sample seems to be larger (are there more contributions?) than the same peaks for the other samples. How do the authors explain this difference? The fit parameters (position, area and FWHM) for all the samples should be reported in a table and the results should be discussed in the text. The same (a Table with the fitting parameters) should be done for the XPS spectra reported in Fig.S2 and Fig.S3. Indeed the peak positions and the FWHM are important in order to understand the chemical bindings, for example the distance between the two peaks relative to La3d5/2 signal is generally used to identify whether La is bound to O or to OH or to CO3 (La2O3, La(OH3)), La2(CO3)3).

Ans. In the Supplementary file, in Fig.S1b, both the Ti2p3/2 and the Ti2p1/2 signals obtained for LT sample seems to be larger, but that is not the case. The fit parameters (position, area and FWHM) of different peaks are now tabulated in Table 3, Table S1 and Table S2. The distance between two peaks (La3d 5/2 and La 3d3/2) corresponds to bonding of La with O.

Comment 5: Page5, line 142 “(Fig.3c)” and “(Fig.3d)” are inverted.

Ans. The Fig. 3c and Fig. 3d are not inverted.

Reviewer 3 Report

The manuscript deals with the synthesis of NPs from TiO2 doped with different amounts of alkali metals like Na and La. The as prepared NPs are characterized  with respect to BET, STEM, TEM/HRTEM, XPS and XRD. Adsorption studies reveal the good sorption or binding properties of LNT4 NP for a cationic dye removal from aqueous solution. 

The manuscript deserves publication only after extensive editing and revision. 

1. The Introduction needs to be revised and following references need to be cited:

 Chemical Engineering Journal 180, 81-90 (2012)

 Rsc Advances 2 (16), 6380-6388 (2012)

 Advances in colloid and interface science 193, 24-34 (2013)

 Critical Reviews in Environmental Science and Technology 45 (6), 613-668 (2015)

Materials Science and Engineering: C 31 (5), 1062-1067 (2011)

Journal of colloid and interface science 493, 228-240 (2017)

2. The authors should conduct isotherm studies too as such studies reveal the binding mechanism of dyes onto the NPs

3. Lower pHIEP values being responsible for higher adsorption properties for LNT4 is not correct as per the studies conducted. Hence, such statement need to be removed from Conclusion.

4. The NPs have tremendous potential to adsorb pollutants from water. The authors should access the adsorptive characteristics of NPs by using higher doses of cationic dye.

5. During the discussion of the results on the batch tests, results should be backed and supported by literature data.

5. Lastly, a comparative analysis need to be performed to assess the performance of NPs vis-a vis other reported adsorbents.

Author Response

Response to reviewer 3

We are thankful to reviewer for reviewing our manuscript.

Comment 1:  The Introduction needs to be revised and following references need to be cited:

 Chemical Engineering Journal 180, 81-90 (2012)

 RSC Advances 2 (16), 6380-6388 (2012)

 Advances in colloid and interface science 193, 24-34 (2013)

 Critical Reviews in Environmental Science and Technology 45 (6), 613-668 (2015)

Materials Science and Engineering: C 31 (5), 1062-1067 (2011)

Journal of colloid and interface science 493, 228-240 (2017)

Ans. All the suggested references (except one) cited as ref. 8, 11, 16-18 in the revised manuscript. One reference Materials Science and Engineering: C 31 (5), 1062-1067 (2011) is not cited as it describes the degradation of dye instead of adsorption.

Comment 2: The authors should conduct isotherm studies too as such studies reveal the binding mechanism of dyes onto the NPs.

Ans. The authors are thankful for this valuable suggestion. The adsorption isotherm is investigated and results are described in the revised manuscript.

Comment 3:  Lower pHIEP values being responsible for higher adsorption properties for LNT4 is not correct as per the studies conducted. Hence, such statement needs to be removed from Conclusion.

Ans. As per reviewer suggestion the mentioned comment is removed in the revised manuscript.

Comment 4:  The NPs have tremendous potential to adsorb pollutants from water. The authors should access the adsorptive characteristics of NPs by using higher doses of cationic dye.

Ans. The effect of higher doses of cationic dye is already investigated in adsorption studies (Effect of dye concentration, Fig. 6(e)) and optimized values are obtained.

Comment 5: During the discussion of the results on the batch tests, results should be backed and supported by literature data.

Ans. The authors have taken care of this comment in the revised manuscript.

Comment 6: Lastly, a comparative analysis need to be performed to assess the performance of NPs with other reported adsorbents.

Ans. On searching literature, we were unable to find adsorption of MB dye on NPs. Therefore, we are unable to do comparative analysis.

Round  2

Reviewer 1 Report

Dear Editor,

My major concerns have been properly addressed. Now the paper is publishable.

Author Response

Response to reviewer 1

We are thankful to reviewer for reviewing our manuscript.

Reviewer 2 Report

The authors have answered all the questions and comments I have raised, therefore the manuscript can be published.

Only two comments to their answers:

1. Ans. The absorbance spectra of MB at different pH is shown in Fig. S4 of supplementary file. There is no change in peak position or shape of absorbance peak when pH is varied. The concentration of dye is calculated using absorbance peak at 664 nm.

The absorbance spectra at pH 7 and 9 are not visible. Please change the graph to make them visible, otherwise the graph is not useful.
2. Ans. The Fig. 3c and Fig. 3d are not inverted.

Please, check again: which is the right one? The text at page 5 line 144: “LNT4 (Fig. 3 (c)) as compared to that of PT (Fig. 3(d))” or the text below fig. 3: “HRTEM images of (c) PT and (d) LNT4”?

Author Response

Response to reviewer 2

We are thankful to reviewer for reviewing our manuscript.

Comment 1 The absorbance spectra at pH 7 and 9 are not visible. Please change the graph to make them visible, otherwise the graph is not useful.

Ans. The absorbance spectra at pH 7 and 9 are not clearly visible due to near complete removal of MB dye at these pH. However, for better clarity the absorbance spectra at pH 7 and 9 are magnified and shown separately as Fig. S4 (b) in revised supplementary file.

Comment 2 Please, check again: which is the right one? The text at page 5 line 144: “LNT4 (Fig. 3 (c)) as compared to that of PT (Fig. 3(d))” or the text below fig. 3: “HRTEM images of (c) PT and (d) LNT4”?

Ans. We apologize for not grasping your suggestion earlier. The text part at page 6 line 176 has been accordingly corrected in revised manuscript.